# Ticks and Rickettsiae Associated with Wild Animals Sold in Bush Meat Markets in Cameroon

**DOI:** 10.3390/pathogens12020348

**Published:** 2023-02-19

**Authors:** Archile Paguem, Kingsley Manchang, Pierre Kamtsap, Alfons Renz, Sabine Schaper, Gerhard Dobler, Deon K. Bakkes, Lidia Chitimia-Dobler

**Affiliations:** 1Department of Comparative Zoology, Institute of Evolution and Ecology, University of Tübingen, 72076 Tübingen, Germany; 2Department of Veterinary Medicine, Faculty of Agriculture and Veterinary Medicine, The University of Buea, Buea P.O. Box 63, Cameroon; 3Bangangte Multipurpose Research Station, Bangangte P.O. Box 222, Cameroon; 4Bundeswehr Institute of Microbiology, Neuherbergstrasse 11, 80937 Munich, Germany; 5Department of Parasitology, Institute of Zoology, University of Hohenheim, 70599 Stuttgart, Germany; 6Gertrud Theiler Tick Museum, Onderstepoort Veterinary Research, Agricultural Research Council, Pretoria 0110, South Africa

**Keywords:** ticks, tick-borne pathogens, bush meat, Cameroon

## Abstract

Ticks are obligate blood-sucking parasites of wild animals and transmit many zoonotic microorganisms that can spread to domesticated animals and then to humans. In Cameroon, little is known about tick diversity among wildlife, especially for animals which are hunted for human consumption. Therefore, this survey was undertaken to investigate tick and *Rickettsia* species diversity parasitizing the wild animals sold in bush meat markets in Cameroon. In total, 686 ticks were collected and identified to the species level based on morphology, and some were genetically analyzed using the 16S rRNA gene. Eighteen tick species belonging to five genera were identified: *Amblyomma* spp. (*Amblyomma compressum*, *Amblyomma flavomaculatum*, and *Amblyomma variegatum*), *Haemaphysalis* spp. (*Haemaphysalis camicasi*, *Haemaphysalis houyi*, *Haemaphysalis leachi*, and *Haemaphysalis parmata*), *Hyalomma* spp. (*Hyalomma nitidum*, *Hyalomma rufipes*, and *Hyalomma truncatum*), *Ixodes* spp. (*Ixodes rasus* and *Ixodes moreli*), and *Rhipicephalus* spp. (*Rhipicephalus guilhoni*, *Rhipicephalus moucheti*, *Rhipicephalus muhsamae*, *Rhipicephalus microplus*, *Rhipicephalus camicasi*, and *Rhipicephalus linnaei*). In terms of *Rickettsia* important for public health, two *Rickettsia* spp., namely *Rickettsia aeschlimannii* and *Rickettsia africae*, were detected in *Hyalomma* spp. and *Amblyomma* spp., respectively. Distinct tick–pathogen patterns were present for divergent sequences of *R. africae* associated with exclusively *A. variegatum* vectors (type strain) versus vectors comprising *A. compressum*, *A. flavomaculatum*, and *A. variegatum.* This suggests possible effects of vector species population dynamics on pathogen population circulation dynamics. Furthermore, *Candidatus* Rickettsia africaustralis was detected for the first time in Cameroon in *I. rasus*. This study highlights the high diversity of ticks among wildlife sold in bush meat markets in Cameroon.

## 1. Introduction

Ticks are obligate blood-sucking arthropods which parasitize every class of terrestrial vertebrate, including mammals, birds, reptiles, and even amphibians [1]. Ticks are primarily parasites of wild animals, with only ~10% of species feeding on domestic animals [2,3]. During feeding on their hosts, they can cause various clinical manifestations including tissue injury, body paralysis, and anemia during massive infestations [4]. Many tick species are known reservoirs and vectors for a multitude of pathogens such as helminths, protozoa, bacteria, and viruses [3]. Wild animals play a significant role as reservoirs for many tick-borne infections, which can be easily spread to domestic animals and then to humans via tick infestations [5]. The emergence and resurgence of several tick-borne diseases continues to pose increasing public health concerns and economic impacts [6,7,8,9,10]. Although the vectorial significance of selected tick species infesting domesticated animals and humans is well-established, there are still many unknowns with respect to genetic diversity, species delimitation, and host distribution, particularly in wildlife [5,11]. 

In Cameroon, 53 ixodid tick species are known from domestic and wild fauna [12,13,14,15]. The majority of studies have focused on ticks of importance for domestic animals and livestock [16,17,18,19]. The most comprehensive study dates back to 1958 [14]. Nevertheless, information on tick species that infest wild animals in Cameroon remains scant. This lack of knowledge is compounded by the rapid decline and/or extinction of many wild species in the region, with associated losses among their highly specialized obligate hematophagous tick parasites [20,21]. *Amblyomma rhinocerotis*, a tick specialized to feed on rhinoceros (*Diceros bicornis*), is already extinct in Cameroon due to growing anthropogenic impacts [22]. To conserve poorly known but highly threatened tick species, which may form the basis of future discoveries yet unknown, a greater understanding of their distribution, ecology, and biology is required [20].

Six ixodid genera, *Amblyomma*, *Dermacentor*, *Haemaphysalis*, *Hyalomma*, *Ixodes*, and *Rhipicephalus*, which contain 34 species between them, have been reported from wildlife fauna in Cameroon [14]. Previous studies were done exclusively based on morphology using museum or private collections [12,13,14,15]. However, morphology alone can prove insufficient to differentiate between closely related tick species, especially when the specimens are damaged, engorged, or at immature stages. Molecular data applied in a phylogenetic comparative framework using verifiable and taxonomically confirmed sequence data can serve to improve the taxonomic robustness of biodiversity surveys. The aim of this study was to identify ticks parasitizing wild animals sold in the bush meat markets in Cameroon using morphological and genetic methods and, in addition, the associated *Rickettsia* spp.

## 2. Materials and Methods

### 2.1. Study Area

The study areas were localized in three large agroecological zones of Cameroon: the Sudano-Sahelian zone (AEZ I), the High Guinea Savannah zone (AEZ II), and the Humid Forest zone with bimodal rainfall (AEZ V) (Figure 1). A strong climatic gradient runs from the humid forest in the south up to the dry Sudano-Sahelian in the far northern region. The rainy season in the Guinea Savannah zone is from April to October, whereas in the Sudano-Sahelian zone, it is from June to September. Annual rainfall ranges from 1400 to 1700 mm in the Guinea Savannah zone and from 800 to 1400 mm in the Sudano-Sahelian zone. The Humid Forest zone with bimodal rainfall lies at altitudes ranging between 400 and 1000 m above sea level. This zone is characterized by a humid forest and savannah mosaic. The average temperature in this zone is about 26 °C, and the annual average rainfall is 2456.8 mm.

These regions were selected because of the presence of hunting, butchering, and consumption of wild meat practices. These areas are also considered to be hot spots for zoonotic disease outbreaks [23]. More than 900,000 reptiles, birds, and mammals are sold each year by or to the rural and urban populations, corresponding to around 12,000 tons of terrestrial vertebrates [24]. Mammals represent the largest proportion of hunted animals, followed by reptiles, birds, and amphibians [24]. 

### 2.2. Tick Sampling

A cross-sectional study was conducted between June 2020 to November 2021 in five bush meat markets in five localities, Kaele (10°6′00″ N 14°27′00″ E), Soramboum (7°47′14″ N 15°0′22″ E), Lom Pangar (5°22′17″ N 13°30′45″ E), Mfou (3°57′36″ N 11°55′48″ E), and Ebolowa (2°55′ 012″ N11°09′00″ E), in the regions of far-northern, northern, eastern, central, and southern Cameroon, respectively (Figure 1). 

Wild animals sold in the bush meat markets of the above-mentioned places were selected randomly and/or conveniently after explaining the aims of the study and upon agreement of the sellers. Most animals were dead and presented individually on a table or on the ground, together with other farming products. No financial incentive was given. Domestic animals are rarely seen in such places, except live chickens, which are kept apart from the bush meat. Examination of each animal was done at the market with as little disturbance as possible. Selected animals were thoroughly examined for attached ticks.

### 2.3. Morphological and Molecular Identification of the Tick

All ticks were identified to the species level using morphological characteristics described by [25,26,27,28,29,30,31].

Total DNA was extracted using the MagNA Pure LC RNA/DNA Kit (Roche, Mannheim, Germany) in a MagNA Pure LC instrument (Roche) according to the manufacturer’s instructions. DNA was extracted from individual ticks (if a single tick was found on a single host) or pools (2–10 ticks per pool, if ticks belonged to the same species and developmental stage and were collected from the same animal). The extracted total DNA was stored at −80 °C until PCR analysis. The 16S rRNA gene was amplified using polymerase chain reaction (PCR) protocols as described by Halos et al. [32].

### 2.4. Sequencing and Phylogenetic Analysis

All obtained 16S rRNA gene sequences for ticks from this study (GenBank accessions: ON318327-ON318387), as well as additional data from GenBank, were compiled into two datasets to represent Rhipicephalinae (*Rhipicephalus* and *Hyalomma*: 117 sequences) and non-Rhipicephalinae (*Ixodes*, *Haemaphysalis*, and *Amblyomma*: 97 sequences). Sequences from GenBank were chosen to encompass the range of *Rhipicephalus*, *Hyalomma*, *Amblyomma*, *Haemaphysalis*, and *Ixodes* species that occur in Afrotropical areas as well as closely related species to counterfactually confirm species identification based on morphology. The validity of species identification using sequences from GenBank follows from recent studies that include large-scale taxonomic investigations to verify species identity via phylogenetic analysis and correlated morphology [33,34,35,36,37]. The prevalence of misidentified tick species among sequence data in GenBank is a growing problem that can only be addressed by large-scale comparative taxonomic studies. Sequence data were aligned using MAFFT (Q-INS-i, 200PAM/k = 2; Gap opening penalty, 1.53) [38]. The optimal nucleotide substitution model was selected using BIC calculations in W-IQTREE [39] and was determined as TPM2+F+G4 (Rhipicephalinae) and K3Pu+F+I+G4 (non-Rhipicephalinae). Maximum likelihood analysis was performed in MEGA v7.0.14 [40] with 1000 bootstraps as well as with calculation of pairwise p-distances. Average p-distances between conspecific sequences from GenBank and collected samples were calculated to determine species identification validity according to the generally accepted threshold of 5% or greater sequence divergence between species [33,35,41,42,43].

*Rickettsia* spp. detection was done using real-time PCR to amplify a fragment of the *gltA* gene for screening. In this first step, 114 *Rickettsia* spp. positive pools were detected. All positive pools with CT below 35 were further investigated by amplifying the 23S-5S intergenic space gene using a conventional PCR. Sequence data for *Rickettsia* 23S-5S (ON333673-ON333740) obtained from this study were compiled with reference data from Genbank and aligned using MAFFT (Q-INS-i, 200PAM/k = 2; Gap opening penalty, 1.53) [38]. The optimal nucleotide substitution model was selected using BIC calculations in W-IQTREE [39] and was determined as HKY+F. Maximum likelihood analysis was performed in MEGA v7.0.14 [40] with 1000 bootstraps.

## 3. Results

### 3.1. Tick Collection and Identification

In total, 686 tick specimens, including 392 males, 179 females, 101 nymphs, and 14 larvae were collected from 166 out of 2000 wild animals examined. Infested wild animals belonged to thirteen species: *Phataginus tricuspis* (white-bellied pangolin, *n* = 48), *Varanus niloticus* (Nile monitor, *n* = 31), *Atelerix albiventris* (four-toed hedgehog, *n* = 25), *Cephalophus rufilatus* (red flanked duikers, *n* = 15), antelopes (*n* = 15), *Lepus victoriae* (African savanna hare, *n* = 9), *Cercopithecus* sp. (monkey, *n* = 7), *Caracal aurata* (African golden cat, *n* = 5), *Mastomys natalensis* (rodent, *n* = 4), *Civettictis civetta* (African civet, *n* = 3), *Phacochoerus africanus* (warthog, *n* = 2), *Python sebae* (African rock python, *n* = 1), and *Atherurus africanus* (African brush-tailed porcupine, *n* = 1). Based on morphological characteristics and 16S rDNA sequencing (Figure 2 and Figure 3), 18 different tick species belonging to *Amblyomma*, *Haemaphysalis*, *Hyalomma*, *Ixodes*, and *Rhipicephalus* genera were identified.

Overall, *A. compressum* was the most frequently identified species (286/686, 41.7%), followed by *A. variegatum* (126/686, 18.36%), *Ha. houyi* (76/686, 11.07%), *Ha. camicasi* (58/686, 8.45%), *A. flavomaculatum* (39/686, 5.7%), and *Rh. microplus* (32/686, 4.7%). The remaining 12 species were found in proportions ranging from 0.1% to 1.6%. *Amblyomma compressum* was also the only species in which all life stages were collected. Adults and nymphs were collected for five out of the 18 species: *A. flavomaculatum*, *A. variegatum*, *Hy. truncatum*, *Rh. moucheti*, and *I. rasus*. Five collected species were represented by adults (males and females): *Ha. camicasi*, *Ha. houyi*, *Ha. leachi*, *Hy. rufipes*, and *Rh. guilhoni*; *Ha. parmata*, *I. moreli*, and *Rh. microplus* were represented by only females, and *Rh. camicasi* was represented by only one male. *Hyalomma nitidum* and *Rh. linnaei* were less often collected, having only one nymph for each species. Only the immature stages of *Rh. muhsamae* were collected. An overview of the tick species, life stages, associated hosts, and collection locality data is summarized in Table 1.

### 3.2. Tick Species and Hosts

Associations of tick species with different hosts are presented in Table 2. Of the 18 tick species infesting 13 species of wild animal, only two species, namely *A. compressum* and *Ha. camicasi*, were common on eight and seven different host species, respectively. Most *A. compressum* (251/286) and *A. flavomaculatum* (17/39) were found on pangolins, whereas *A. variegatum* (99/126) was found on Nile monitors. *Rhipicephalus muhsamae* (9/10) was associated with rats, with only one nymph found on Nile monitors. Ten out of eighteen species were recorded from Nile monitors, whereas only one species, *A. compressum*, was recorded from golden cats (*n* = 2). Four-toed hedgehogs were infested with eight different tick species. Five species (*Hy. nitidum*, *Hy. rufipes*, *I. moreli*, *Rh. linnaei*, and *Rh. camicasi*) were recorded from a single host species each (Table 2). *Rhipicephalus microplus* was recorded for the first time on four-toed hedgehogs (31/32) and Nile monitors (1/32).

### 3.3. Phylogenetic Analysis of the Partial 16S rRNA Gene

Phylogenetic trees are shown in Figure 2 and Figure 3. *Amblyomma flavomaculatum*, *Ha. camicasi*, *Ha. houyi*, *I. moreli*, and *Rh. moucheti* were sequenced for the first time. Additionally, one fully engorged *Hyalomma* female that was difficult to identify morphologically was confirmed phylogenetically as *Hy. truncatum*. Six *I. rasus*, three *Rh. guilhoni*, and one *Rh. muhsamae* collected from our study areas were sequenced and phylogenetically confirmed as well. 

### 3.4. Tick Species Distribution in the Three Agroecological Zones

The relative abundance and distribution of each tick species for each agroecological zone is described in Table 3. 

In the Sudano-Sahelian zone, eight tick species were recorded: *A. flavomaculatum*, *A. variegatum*, *Ha. houyi*, *Hy. rufipes*, *Hy. truncatum*, *Rh. microplus*, *Rh. muhsamae*, and *Rh. linnaei*. *Rhipicephalus microplus* is reported for the first time in this region and host. In the High Guinea Savannah zone, six tick species (*Ha. camicasi*, *Ha. houyi*, *Hy. nitidum*, *Rh. moucheti*, *Rh. guilhoni*, and *Rh. camicasi*) were collected from seven hosts. *Haemaphysalis* species were associated with seven hosts. *Rh. moucheti* is reported for the first time in this region. In the Humid Forest zone with bimodal rainfall, seven tick species were collected: *A. compressum*, *A. flavomaculatum*, *Ha. camicasi*, *Ha. parmata*, *Ha. leachi*, *I. rasus*, and *I. moreli*. 

In three cases, tick species were found in two different agroecological zones. *Amblyomma flavomaculatum* was found in the Sudano-Sahelian and Humid Forest with bimodal rainfall zones, whereas *Ha. camicasi* was found in the High Guinea Savannah and Humid Forest with bimodal rainfall zones, and *Ha. houyi* was found in the Sudano-Sahelian and High Guinea Savannah zones.

### 3.5. Rickettsia spp. Detection

In total, 66 (57.9%) out of 114 positives from screening were sequenced.

Phylogenetic analysis showed three different *Rickettsia* species (Figure 4) circulating in ticks collected from wild animals, namely *Rickettsia africae*, *Rickettsia aeschlimannii*, and *Candidatus* Rickettsia africaustralis. *Rickettsia africae* (61, 92.4% of the 66 sequenced samples) was detected in all *Amblyomma* spp., with a high infestation rate in *A. compressum* (47/61, 77% of all *Amblyomma* spp.) but less in *A. variegatum* (10/61, 16.4%) and *A. flavomaculatum* (4/61, 6.55%). Moreover, distinct tick–pathogen patterns were present among divergent sequences of *R. africae* associated with exclusively *A. variegatum* (type strain) (within-clade sequence dissimilarity: ~0.008%) versus *A. compressum*, *A. flavomaculatum*, and *A. variegatum* (within-clade sequence dissimilarity: ~0%) (between-clade sequence dissimilarity: ~0.006% mean pairwise dissimilarity). *Rickettsia aeschlimannii* was detected in two *Hy. rufipes* and one *Hy. truncatum*. *Candidatus* Rickettsia africaustralis was detected for the first time in Cameroon in two *I. rasus* females, both of which were collected from pangolins.

## 4. Discussion

Limited studies focused on ticks parasitizing wild animals in Central Africa and their role as carriers of rickettsial bacteria. Traditional hunting and bush meat markets play a central role in contact between animals and humans and the spread of diseases from rural to urban populations. The selection of animals sold at such markets does not necessarily reflect natural abundances of the respective animal fauna. It is, however, useful for studying the human–vector–parasite interface and the diversity of tick, pathogen, and reservoir species involved. Due to the nature of bush markets, potential limitations include ticks leaving their hosts before examination or moving from one animal to another, though the prevalence of such events are generally rare and can be controlled for with large sample sizes. Additionally, vendors usually offer only one of any game species, further limiting the possibility of cross-contamination.

### 4.1. Tick Species and Hosts

In total, eighteen tick species belonging to *Amblyomma*, *Haemaphysalis*, *Hyalomma*, *Ixodes*, and *Rhipicephalus* genera were collected from thirteen wild animal species.

Three *Amblyomma* species were recorded from wild animals, with *A. compressum* and *A. flavomaculatum* predominantly found on pangolins (noting the endangered status of pangolins and it being illegal to trade these animals in the markets in Cameroon since April, 2020), but also on Nile monitors and monkeys, whereas *A. variegatum* were found on Nile monitors and Four-toed hedgehogs in the Sudano-Sahelian zone. *Amblyomma compressum* is known as an exclusive ectoparasite of three African pangolin species [28]. However, in our study, *A. compressum* was collected from seven unexpected hosts, namely warthogs, Nile monitors, African brush-tailed porcupines, monkeys, African gold cats, African civets, and antelopes. Adults and immatures were collected from pangolins, whereas adults and nymphs were collected from African civets, monkeys, and warthogs, with only adults collected from the remaining hosts. This finding suggests that *A. compressum* is not very host-specific, and that cohabitation of pangolins with unexpected hosts can lead to infestations of the pangolin tick. *Amblyomma flavomaculatum* is known as a reptile tick, especially on Nile monitors and Savannah monitors (*Varanus exanthematicus*) [14,44], and it was previously reported from both hosts in Cameroon [14]. However, in the present study, *A. flavomaculatum* was collected from four hosts other than Nile monitors, namely, pangolins, four-toed hedgehogs, monkeys, and warthogs. It seems the pangolin is also an important host, as 17 specimens were collected from this host, while 16 specimens were collected from Nile monitors. Adults and two nymphs were collected from Nile monitors, while only adults were collected from unexpected hosts.

Conversely, *A. variegatum* is a generalist parasitizing a large spectrum of vertebrates including domestic animals, wild animals, and humans [14]. Seven *Amblyomma* species were reported from wildlife in Cameroon [14], namely *A. cohaerens* and *A. splendidum* from buffalo (*Syncerus coffer*), *A. tholloni* from African elephants (*Loxodonta africana*), *A. paulopunctatum* from domestic and wild suids (warthogs, pigs, etc.), *A. nuttalli* from tortoises, *A. exornatum* and *A. flavomaculatum* from Nile monitors (*Varanus niloticus* and *V. exanthematicus*), and *A. latum* as well as *A. transversale* from snakes (*Naja melanoleuca*, *Python sebae*, and *Python regius*). 

In the genus *Rhipicephalus*, six species were recorded on Nile monitor, four-toed hedgehog, African savanna hare, red flanked duiker, monkey, and rodent hosts. *Rhipicephalus microplus*, *Rh. muhsamae*, and *Rh. linnaei* were exclusively found in Sudano-Sahelian areas, whereas *Rh. guilhoni*, *Rh. moucheti*, and *Rh. camicasi* were found in High Guinea Savannah areas. *Rhipicephalus microplus* is currently expanding its geographical and host range, having been previously reported on cattle in Humid Forest and High Guinea Savannah zones, but in our study, it was found in the Sudano-Sahelian dry savannah and steppes on Nile monitors and four-toed hedgehogs for the first time. This is important to note, as *Rhipicephalus* are involved in the transmission of viral, bacterial, and protozoan agents to domestic and wild animals as well as humans.

From the genus *Hyalomma*, three species were recorded: *Hy. rufipes*, *Hy. truncatum*, and *Hy. nitidum*. Adults of *Hy. rufipes* and *Hy. truncatum* were found on four-toed hedgehogs, whereas immatures of *Hy. truncatum* and *Hy. nitidum* were found on Nile monitors and red flanked duikers, respectively. The genus is well-represented in Cameroon by five species: *Hy. impeltatum*, *Hy. impressum*, *Hy. rufipes*, *Hy. truncatum*, and *Hy. nitidum*, which are parasites of ruminants, birds, small mammals, and often humans [14]. They are involved in the transmission of *R. aeschlimannii* and Crimean–Congo hemorrhagic fever to domestic and wild animals as well as humans.

The genus *Haemaphysalis* is represented in Cameroon by at least six species: *Haemaphysalis aciculifer*, *Ha. hoodi*, *Ha. houyi*, *Ha. leachi*, *Ha. muhsamae*, and *Ha. parmata* [14]. Most of them feed on primitive, medium-sized, and small-sized mammals as well as birds. In this study, four species, *Ha. camicasi*, *Ha. houyi*, *Ha. leachi*, and *Ha. parmata*, were recorded. The paraphyly between *Ha. leachi* lineages between Chad and Cameroon is puzzling, but it may indicate an unknown divergence (species or population) or misidentification associated with sequence data. The *Ha. leachi* complex remains a taxonomic challenge to date [45].

Ticks of the genus *Ixodes* with seven species are poorly studied in Cameroon. In this highly specialized group of ticks, many immature stages are not even known. Two species were reported: *I. rasus* and *I. moreli*. 

From the 18 species identified in this study, nine species are known to parasitize humans: *A. variegatum*, *Ha. leachi*, *Hy. rufipes*, *Hy. truncatum*, *I. rasus*, *Rh. microplus*, *Rh. guilhoni*, *Rh. muhsamae*, and *Rh. linnaei* (previously known as *Rh. sanguineus* s.l.)

### 4.2. Phylogenetic Analysis of the Partial 16S rRNA Gene

Five tick species collected in the present study, *A. flavomaculatum*, *Ha. camicasi*, *Ha. houyi*, *I. moreli*, and *Rh. moucheti*, were sequenced for the first time. The phylogenetic analysis and sequences submitted to GenBank are relevant for future studies. Additionally, sequences confirmed morphological determinations for fully engorged specimens, larvae, and/or nymphs as well as for species collected from unexpected wild animal hosts like *Rh. microplus* and *Rh. linneai*. In total, 214 specimens were sequenced and included in the genetic analysis. Phylogenetic analysis confirmed the presence of *Rh. camicasi* in Cameroon, in sympatry with *Rh. guilhoni*. Both species belong to the *Rhipicephalus sanguineus* group and have a similar morphology, but they have different geographic distributions [29]. 

### 4.3. Rickettsia spp. Detection

The genus *Amblyomma* is involved in the transmission of viral, bacterial, and protozoan agents to domestic and wild animals as well as humans. These include *Ehrlichia ruminantium* (the agent of heartwater), *Theileria mutans*, and *Rickettsia africae* (the agent of African tick-bite fever) [46]. There are many reports about *A. variegatum* carrying *R. africae* from different African countries such as Sudan [47,48] and Ethiopia [49]. In the present study, *R. africae* was detected in *A. compressum*, *A. variegatum*, and *A. flavomaculatum* and was sequenced from 61 tests (92.4% of the 66 sequenced samples). Phylogenetic analysis showed three clusters of *R. africae* detected in *A. variegatum* but only one cluster for *A. compressum* (primarily), *A. variegatum*, and *A. flavomaculatum*. The mean sequence divergence between *A. variegatum*-exclusive sequences and the wider vector associated clade (albeit primarily associated with *A. compressum*) was 0.006%. This may be related to the particulars of distinct tick species and their population dynamics, and it may not be directly linked with tick–host–pathogen biochemistry. More precisely, the high phylogenetic diversity of *R. africae* in *A. variegatum* may be due to the wide host range of *A. variegatum* that adds complexity and redundancy to the patterns of population dynamics among *R. africae* and *A. variegatum* populations. Furthermore, this suggests possible effects of vector species population dynamics on pathogen population circulation dynamics in the context of host population dynamics (a triple-layered system), which may drive evolutionary divergences in *Rickettsia* populations, strains, and species, or it may potentially cause differential pathogenicities to emerge over significant evolutionary time. *Rickettsia aeschlimannii* is associated with the *Hyalomma* species, especially with *Hy. rufipes* in Africa [47,48] as well as in European countries where *Hy. rufipes* is introduced by migratory birds as nymphs [50,51]. *Rickettsia aeschlimannii* was detected in two *Hy. rufipes* and one *Hy. truncatum*. The low number of *R. aeschlimannii* reported here is possibly due to the reduced number of *Hyalomma* species on wild animals. 

*Candidatus* Rickettsia africaustralis was detected for the first time in Cameroon in two *I. rasus* females, both collected from pangolins. This *Rickettsia* species was detected for the first time in small mammal tissue from South Africa [52]. *Ixodes rasus* may prove to be a vector and reservoir for *Candidatus* R. africaustralis, as in other related *Ixodes* spp. Further investigations are necessary regarding the pathogenicity and vector competency for this new *Rickettsia* species. 

## 5. Conclusions

The present study was carried out to determine the tick species that infest wild animals in Cameroon. As far as is known, this is the first tick survey of wild animals comprising tick species among five genera from well-defined hosts and agroecological zones. Ticks were identified morphologically and further confirmed or characterized genetically. In total, eighteen tick species belonging to *Amblyomma*, *Haemaphysalis*, *Hyalomma*, *Ixodes*, and *Rhipicephalus* genera were collected from thirteen wild animal species. The most common bush meat species were the brush-tailed porcupine (*Atherurus africanus*), duiker (*Cephalophus monticola*), pangolin (*Phataginus tricuspis*), and Nile monitor (*Varanus niloticus*) [24]. Interactions between wild hosts, vectors, and pathogens are important to understand potential epidemiological risks and drivers.

## Figures and Tables

**Figure 1 pathogens-12-00348-f001:**
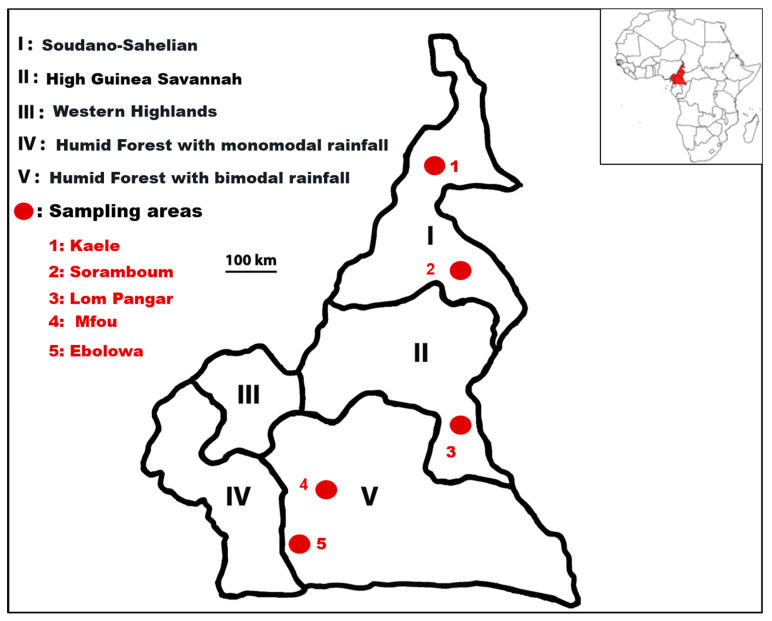
Map of the study area. Geographic map showing five agroecological zones of Cameroon (based on information from the Institute of Agricultural Research for Development, IRAD, 2009). The sampling areas (red circle) were located in the climate zones High Guinea Savannah, Sudano-Sahelian, and Humid Forest with bimodal rainfall.

**Figure 2 pathogens-12-00348-f002:**
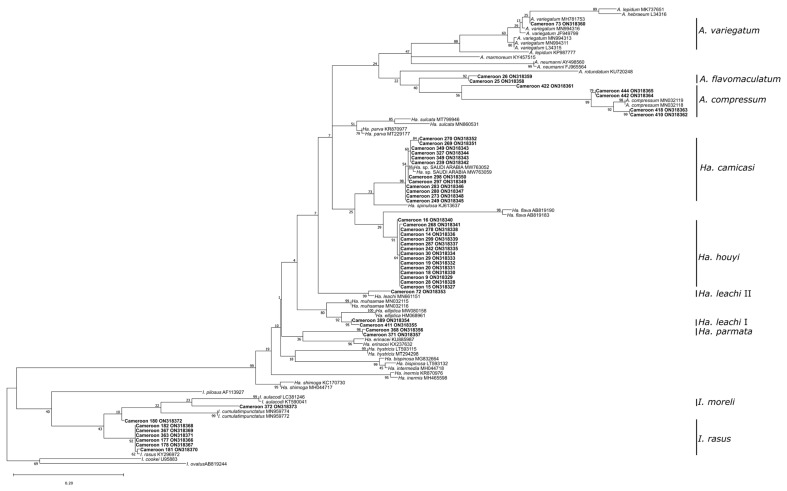
Maximum likelihood phylogenetic analysis of 16S rDNA sequences for non-Rhipicephalinae ticks (*Ixodes, Haemaphysalis,* and *Amblyomma)* using a K3Pu+F+I+G4 nucleotide substitution model. Indicated are sample names and species/lineages collected as well as Genbank accession numbers and bootstrap support values. Bolded samples refer to sequences generated in this study.

**Figure 3 pathogens-12-00348-f003:**
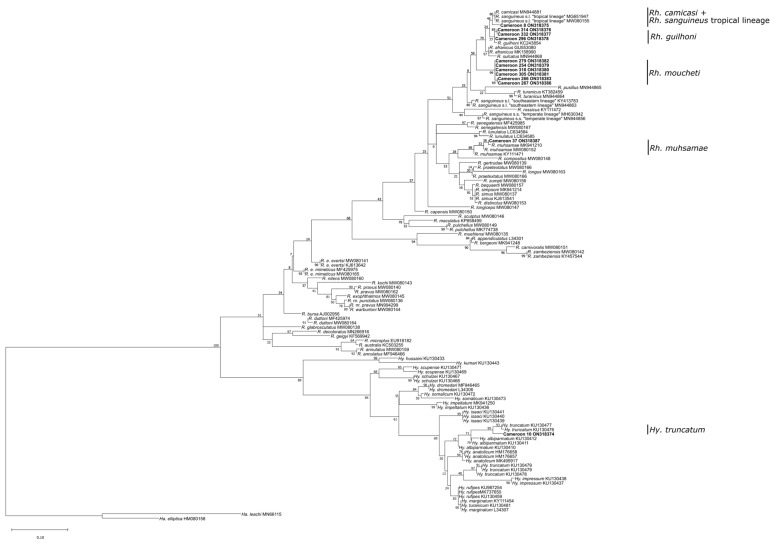
Maximum likelihood phylogenetic analysis of 16S rDNA sequences for Rhipicephalinae ticks (*Rhipicephalus* and *Hyalomma*) using a TPM2+F+G4 nucleotide substitution model. Indicated are sample names and species/lineages collected as well as Genbank accession numbers and bootstrap support values. Bolded samples refer to sequences generated in this study.

**Figure 4 pathogens-12-00348-f004:**
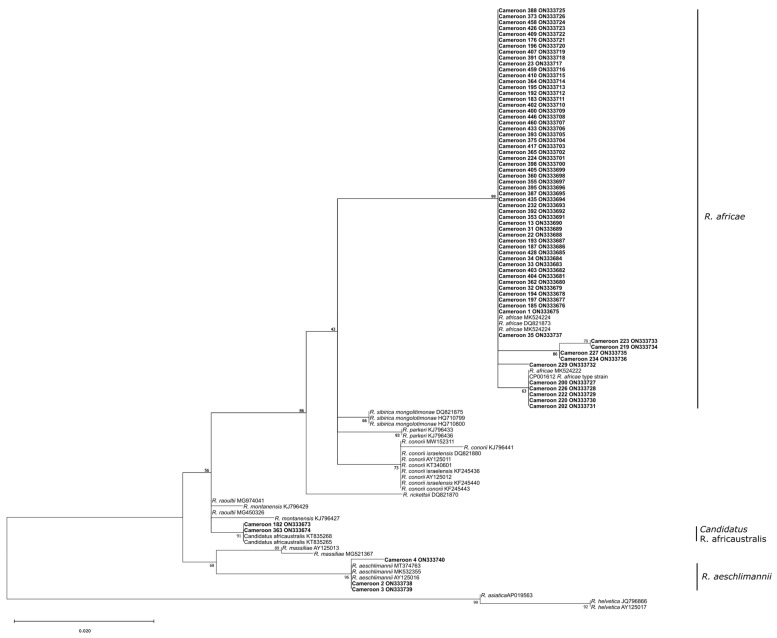
Maximum likelihood phylogenetic analysis of 23S-5S sequences for *Rickettsia* spp. using an HKY+F nucleotide substitution model. Indicated are sample names and species/lineage names as well as Genbank accession numbers and bootstrap support values. Bolded samples refer to sequences generated in this study.

**Table 1 pathogens-12-00348-t001:** Tick species, life stages, associated hosts, collection locality, number, and frequency.

Tick Species	Tick Stage				Host	Location	*n*	Frequency (%)
	Male	Female	Nymph	Larva				
*Amblyomma compressum*	175	46	53	12	White-bellied pangolins, Warthogs, Nile monitors, African brush-tailed porcupines, Monkeys, African golden cat, Antelopes, African civets	Ebolowa, Mfou	286	41.8
*Amblyomma flavomaculatum*	32	5	2		Nile monitors, Four-toed hedgehogs, White-bellied pangolins, Monkey, Warthog	Kaele, Ebolowa	39	5.7
*Amblyomma variegatum*	90	6	30		Nile monitor, Four-toed hedgehog	Kaele	126	18.4
*Haemaphysalis camicasi*	25	33			Red flanked duikers, Nile monitors, Monkeys, Hares, Four-toed hedgehog, African rock pythons, Antelopes	Ebolowa, Lom Pangar	58	8.5
*Haemaphysalis houyi*	43	33			Nile monitors, Red flanked duikers, Monkeys, African savanna hares, Four-toed hedgehogs, Antelopes, African rock python	Kaele, Lom Pangar	76	10.9
*Haemaphysalis parmata*		2			Nile monitor, Antelope	Ebolowa	2	0.3
*Haemapysalis leachi*	11	6			White-bellied pangolins, African civets	Ebolowa	17	2.5
*Hyalomma nitidum*			1		Red flanked duiker	Lom Pangar	1	0.1
*Hyalomma rufipes*	4	1			Four-toed hedgehog	Kaele	5	0.7
*Hyalomma truncatum*	1	1	4		Four-toed hedgehogs, Nile monitor	Kaele	6	0.9
*Ixodes moreli*		1			Antelope	Ebolowa	1	0.1
*Ixodes rasus*	1	8	1		White-bellied pangolins, Four-toed hedgehog, Warthog, Monkey	Ebolowa	10	1.5
*Rhipicephalis microplus*		32			Four-toed hedgehogs, Nile monitor	Kaele	32	4.7
*Rhipicephalus guilhoni*	2	2			African savanna hare, Four-toed hedgehog, Nile monitor	Lom Pangar	4	0.6
*Rhipicephalus moucheti*	7	3	1		Red flanked duikers, antelope, Monkeys, Four-toed hedgehogs, hares, Nile monitor	Lom Pangar	11	1.6
*Rhipicephalus linnaei*			1		Nile monitor	Kaele	1	0.1
*Rhipicephalus camicasi*	1				Red flanked duiker	Lom Pangar	1	0.1
*Rhipicephalus muhsamae*			8	2	Rats, Nile monitor	Soramboum, Kaele	10	1.5
**Total**							686	100

**Table 2 pathogens-12-00348-t002:** Tick species and the hosts from which they were collected.

Tick Species	Host													
	White-Bellied Pangolin	Warthog	Nile Monitor	African Brush-Tailed Porcupine	Monkey	African Golden Cat	African Civet	Antelope	Four-Toed Hedgehog	Red Flanked Duiker.	African Rock Python	African Savanna Hares	Rodent	Total
*A. compressum*	251	3	16	5	2	2	4	3						286
*A. flavomaculatum*	17	1	16		1				4					39
*A. variegatum*			99						27					126
*Ha. camicasi*			7		15			11	7	9	2	7		58
*Ha. houyi*			2		15			18	15	15	1	10		76
*Ha. parmata*			1					1						2
*Ha. leachi*	3						14							17
*Hy. nitidum*										1				1
*Hy. rufipes*									5					5
*Hy. truncatum*			2						4					6
*I. moreli*								1						1
*I. rasus*	7	1			1				1					10
*Rh. microplus*			1						31					32
*Rh. guilhoni*			2						1			1		4
*Rh. moucheti*			1		3			1	2	2		2		11
*Rh. linnaei*			1											1
*Rh. camicasi*										1				1
*Rh. muhsamae*			1										9	10
Total	278	5	149	5	37	2	18	35	97	28	3	20	9	686

**Table 3 pathogens-12-00348-t003:** Tick species and numbers specific to each agroecological zone.

Genus	Tick Species	No. of Ticks Collected	Sudano-Sahelian Zone	High Guinea Savannah Zone	Humid Forest Zone with Bimodal Rainfall
*Amblyomma*	*A. compressum*	286			286
	*A. flavomaculatum*	39	20		19
	*A. variegatum*	126	126		
*Haemaphysalis*	*Ha. camicasi*	58		56	2
	*Ha. houyi*	76	11	65	
	*Ha. parmata*	2			2
	*Ha. leachi*	17			17
*Hyalomma*	*Hy. nitidum*	1		1	
	*Hy. rufipes*	5	5		
	*Hy. truncatum*	6	6		
*Ixodes*	*I. moreli*	1			1
	*I. rasus*	10			10
*Rhipicephalus*	*Rh. microplus*	32	32		
	*Rh. guilhoni*	4		4	
	*Rh. moucheti*	11		11	
	*Rh. linnaei*	1	1		
	*Rh. camicasi*	1		1	
	*Rh. muhsamae*	10	10		
Total		686	211	138	337

## Data Availability

All of the sequences were deposited in GenBank, and the raw data are in the Bundeswehr Institute of Microbiology.

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
