# Peer review of "Ticks and Rickettsiae Associated with Wild Animals Sold in Bush Meat Markets in Cameroon"

_pathogens, 2023, doi:10.3390/pathogens12020348_

Round 1

Reviewer 1 Report

The manuscript investigated ticks collected from wild animals sold in bush meat markets in three agro-ecological zones of Cameroon. The ticks were identified morphologically and molecularly. Eighteen species of ticks belonging to five genera were collected from 13 species of wild animals. Notably, a lot of them were collected from unexpected hosts. Pathogens were examined by PCR targeting gltA and 23S-5S ITS, and Rickettsia africae was detected in Amblyomma spp., while Rickettsia aeschlimannii was detected in Hyalomma rufipes and Hy. truncatum. Candidatus Rickettsia africaustralis was found in Cameroon for the first time in Ixodes rasus.

Generally, the manuscript is well-written. However, there are some points need to be further addressed, and certain sections are not completed. My major concerns are:

1. considering the journal’s name is “Pathogens”, the manuscript rather focused on tick (vector) itself and described the pathogens they found in the study roughly;

2. different species of captured wild animals likely were kept together in bush meat markets, so one animal found with a tick did not necessarily mean it was the host of the tick;

3. white-bellied pangolin has been listed as Class A on the Cameroon protected species list since April, 2020, and any capture, detention, and trade of live or dead white-bellied pangolin are prohibited at the national level. I wonder if it is ethical to perform tick collection while encountering illegal trade.

Major points:

. Since ticks highly specialized on certain hosts were mentioned in “Introduction” (P2 L50-60), the authors could spent some space discussing the highly specialized ticks they found in the study.

. P2 L73-88: 1. There were no sampling in AEZ III and AEZ IV, why? 2. Would trade of wild animals occur across zones?

. P3 L95-102: 1. What would the authors do when they found endangered species in bush meat markets? Was it ethical to carry on tick collection without reporting to the authorities? 2. Ticks were sampled from live and dead animals. Wouldn’t ticks leave their dead hosts? 3. Did they keep captured wild animals together or close to each other? Were there domestic animals in the same markets? If so, ticks might move from their dead hosts to another animals which were not actually their natural hosts. Or the wild animals might get ticks from the domestic animals appeared in the same markets.

. P4 L154-5: How many ticks/pool of ticks were identified using molecular analysis?

. P5 Figure 2,3: 1. Some values of the branches on the phylogenetic trees were quite low, please explain. 2. Sampling site (country), host, and year should be labeled for each sequences including reference sequences except their species and accession numbers.

. P6 Table 1: 1. “286 41.8” were repeated in the row after Amblyomma compressum’s tick stage. 2. A comma was missing between warthog and monkey in the row of Ixodes rasus’s hosts. 3. Another row should be added for the total capture at the end of the table.

. P7 Table 2: Twelve species of ticks were found on Nile monitor, and 11 of them were also found on other mammals. Both two species of ticks found on African rock python also appeared on other mammals. I wonder if it is common for ticks that a species of tick uses both reptile (ectotherm) and mammal (endotherm) as its hosts.

. P8 L196-8: Not only engorged specimens but also nymph, larvae, and suspected specimens were examined molecularly (P11 L304-6). Please address this, and add the number of specimens tested.

. P9 L205-17: Was there any preference of species of bush meat in different zones? Any difference of the species of wild animals sold in bush meat markets between zones?

. P10 L238-9: Finding a tick on an animal did not necessarily mean the animal was the actual host of the tick. Further analysis should be performed to prove the tick took blood from that animal.

. P10-11 L246-99: Of the 18 ticks identified in the study, which of them would attack human?

. P11 L310-326: The relationship between Rickettsia spp. and their host ticks and animals in different agro-ecological zones can be further discussed.

. P11 L239-30: “Ixodes rasus may prove to be a vector and reservoir for C. Rickettsia africaustralis,” Was there any reference to support this statement?

. P12 L333-40: Please fill out “Author Contributions.”

. P12 L342-51: Please add the Institute Review Board Statement and approval number.

. P12 L352-6: Please fill out “Data availability statement.”

. “References”: Please use the right style for reference.

Minor points:

. Molecular identification of ticks were carried out targeting the 16S rDNA (16S rRNA gene). I would suggest the authors to use the same way, either 16S rDNA or 16S rRNA gene, to describe the targeting gene throughout the manuscript.

. P2 L68-70: “The aim of this study was to identify tick species parasitizing wild animal’s sale in the bush meat market in Cameroon using morphological and genetic methods.” Tick are unlikely to parasitize wild animal’s sale.

. P2 L81-2: “The average temperature is about 26ºC and the annual average rainfall is 2456.8 mm.” Which zone did the description of climate refer to?

. P9 L230: Please italicize “Rickettsia africae.”

. P11 L285: “Red flanked Duiker.” D did not need to be capitalized.

. P11 L315: Italicize “R. africae.”

. P11 L320: Italicize “R. africae” and “A. variegatum.”

. P11 L330: “C. Rickettsia africaustralis” should be Candidatus R. africaustralis.

Reviewer 2 Report

Comments for the authors of the manuscript “Diversity of Ticks Associated With Wild Animals in Cameroon”

The manuscript brings relevant results for publication. Overall, it is well written and structured. I send some suggestions to optimize the presentation of the study:

Abstract- In the objectives it is not mentioned that research will be carried out for rickettsia in ectoparasites, however data on rickettsia infection in ticks are presented in the results. With the presentation of the rickettsia results, I suggest revising the title of the manuscript.

Materials and Methods- Lines 112-114: Why was pooled extraction performed? I suggest detailing why this methodology was used and whether there were any differences in the results (amplification of the analyzed 16S rDNA fragment and detection of rickettsia) between pooled and single samples.

Results- Lines 168-169: The results presented in the text do not correspond to those in Table 1. For example: In the text, A. variegatum (126/686, 18.36%); and in Table 1: Amblyomma variegatum 58/686 (8.5%). Review.

Table 1- The title failed to include the analysis of the frequency of collected species.

Results- Lines 194-195: “A phylogenetic tree was estimated based on a maximum likelihood approach using the partial 16S rRNA” Would not it be methodology?

Results- 3.5. Rickettsia spp. detection: It is not mentioned in the objectives about rickettsia research. And, in the methodology there is no detail about the research of rickettsia in ticks.

Figure 4- I suggest indicating the species of ticks in which the analyzed rickettsiae were detected.

Discussion- Be careful not to repeat the results. This seems to occur a few times throughout the discussion.

Discussion- Line 320: A. variegatum in italics.

Author Contributions, Institutional Review Board Statement, Data Availability Statement: These sections were not filled out properly.

Reviewer 3 Report

Dear authors,

I carefully revised you submitted paper entitled “Diversity of ticks associated with wild animals in Cameroon”. The information the manuscript offer is valuable and a good contribution to the knowledge about tick diversity in Cameroon. Further, I listed you my comments and corrections:

Title

You should include the fact that the ticks were collected on dead animals from bush meat markets to the title.  

Abstract

line 22: please correct 16S rRNA gene (16S not in italic, also see the rest of the manuscript)

line 28: You should not talk about microorganisms important for public health if you only tested for the presence of rickettsial DNA. Please specify it

Material and Methods

line 100: aggregate bush meat market

“Tick sampling” What happened with the sampled animals? Where did you realize the examination of the animals? On the bush meat markets? Please specify that if the animals were dead for some time, that some tick specimens already could have leave the carcass. Therefore, it may be important to mention in which conditions were the sampled animals. The same for the agreement of the sellers. Which were these agreements? Did you pay them? What is about ethical approvals of this study?

Line 118: change 16S rRNA gene sequences

Lines 131/141: Please add the citation to the reference list

Lines 138 ff: Please describe the method that you applied for rickettsial DNA detection in this section and not in the results section (lines 219ff)

Results

lines 194 and 198: 16S rRNA gene

line 219: Rickettsia in italic

lines 219-220: see above, material and methods

line 222: material and methods. Which assay did you use for the detection of 23S-5S IGS?

lines 221-222: All gltA positives pools were also positive to 23S-5S IGS? How did you select the samples for sequencing?

line 230: Rickettsia africae in italic

Discussion

lines 237-244: This is more a conclusion that a discussion. I would send it to the end of the manuscript as “5. Conclusion”

4.1 to 4.3 The discussion is very general. Please try to shorten it and highlight the results of your study. Further, I recommend you the use of scientific names of the animal species instead of the common names as they often change and e.g. monkey (line 260) is not an appropriate description for a host.

line 265: domestic instead of domesticated

line 280: domestic instead of domestics

line 307: it is the Rhipicephalus sanguineus group

line 310: Amblyomma in italics

lines 315 and 320: R. africae in italics

line 320: A. variegatum in italics

line 330: Ca. instead of C.

line 328: It is not a new species if it has the status of Candidatus

Round 2

Reviewer 1 Report

The manuscript reported tick species on wild animals sold in the bushmeat markets in Cameroon. The rickettisal pathogens in ticks were examined. The modified version of manuscript has improved, and most questions have been answered. Yet there are still some mistakes.

1.      The titled has been changed to “Ticks and Rickettsiae Associated With Wild Animals Sold as Bush Meat Market in Cameroon.” It should be “Ticks and Rickettsiae Associated With Wild Animals Sold as Bush Meat in Markets in Cameroon” or “Ticks and Rickettsiae Associated With Wild Animals Sold in Bush Meat Markets in Cameroon.” The wild animals were killed and presented as bushmeat for sale in the bushmeat markets, but they did not became a market to be sold.

2.      P2 L73-5: “The aim of this study was to identify tick species parasitizing wild animal’s sale in the bush meat market in Cameroon using morphological and genetic methods.” Please correct the sentence. Ticks parasitized wild animals sold in the bushmeat markets in Cameroon. They did not parasitize the trade of wild animal. And addition of identification of rickettsiae as one of the aims of the study is suggested.

3.      Figure 2, 3, 4: In order to make it easier for the readers to get the picture of the relationships between these sequences, sampling site (country), host, and year should be labeled for each sequences including reference sequences in addition to their accession numbers, although the information can be found in GenBank. e.g. A. lepidum MK737651_Egypt_camel_2019

4.      I agree that constantly monitoring zoonotic pathogens in ectoparasites on wild animals, especially those used as bushmeat, is critically important. However, I cannot agree on the saying of not interfering with local regulations which allow the traditional hunting of pangolins. If it’s allowed, it would not be banned by the law of the nation. I believe the authors, as educated scholars or foreign researchers dedicated in science with morality, didn’t just want to finish the current study and get it published. The authors have worked in the field in the nation for years not only to investigate into potential tick-borne diseases but to improve people’s living. Instead of standing by, the authors should use their trusty relationship with local inhabitants and their knowledge to let the people know the importance of public health and conservation of endangered species. Local people may not be aware that they have broken the law by trading these animals. Indeed the authorities may not care about the illegal trade happening in the markets, but it did not mean the authors should back off from the issue, too. At least certain description about the endangered status of pangolins and it being illegal to trade these animals in the markets in Cameroon since April, 2020 should be mentioned in the manuscript. Stopping illegal trade of endangered species is not only a way to protect natural resources but a measure to prevent the spread of zoonotic pathogens.

5.      Further English editing is suggested. There are minor grammar mistakes throughout the manuscript.

6.      P12 L380-2: Please add the approval number of the institutional review board.

7.      P12-4 L391-502: Some of the journal names were not abbreviated.

Author Response

Dear Editor,

We completed the revision, considered all the reviewer comments and marked the answer italics.
